# To Kill or to Repel Mosquitoes? Exploring Two Strategies for Protecting Humans and Reducing Vector-Borne Disease Risks by Using Pyrethroids as Spatial Repellents

**DOI:** 10.3390/pathogens10091171

**Published:** 2021-09-11

**Authors:** M. Moreno-Gómez, M. A. Miranda, R. Bueno-Marí

**Affiliations:** 1Research and Development (R&D) Insect Control Department, Henkel Ibérica S.A, Carrer Llacuna 22, 1-1, 08005 Barcelona, Spain; 2Applied Zoology and Animal Conservation Research Group, Biology Department, University of the Balearic Islands, Cra. Valldemossa km 7.5, 07122 Palma de Mallorca, Spain; ma.miranda@uib.es; 3Lokímica S.A., Departamento de Investigación y Desarrollo (I + D), Ronda Auguste y Louis Lumière 23, Nave 10, Parque Tecnológico, 46980 Paterna, Valencia, Spain; rbueno@lokimica.es; 4Área de Parasitología, Departamento de Farmacia y Tecnologia Farmacéutica y Parasitología, Facultad de Farmacia, Universitat de València, Avda. Vicent Andrés Estellés, s/n, 46100 Burjassot, Valencia, Spain

**Keywords:** *Aedes albopictus*, field testing, killing, landing rate count, protection, spatial repellent, transfluthrin

## Abstract

Although control efforts are improving, vector-borne diseases remain a global public health challenge. There is a need to shift vector control paradigms while developing new products and programmes. The importance of modifying vector behaviour has been recognised for decades but has received limited attention from the public health community. This study aims to: (1) explore how the use of spatial repellents at sublethal doses could promote public health worldwide; (2) propose new methods for evaluating insecticides for use by the general public; and (3) identify key issues to address before spatial repellents can be adopted as complementary vector control tools. Two field experiments were performed to assess the effects of an insecticidal compound, the pyrethroid transfluthrin, on *Aedes albopictus* mosquitoes. The first examined levels of human protection, and the second looked at mosquito knockdown and mortality. For the same transfluthrin dose and application method, the percent protection remained high (>80%) at 5 h even though mosquito mortality had declined to zero at 1 h. This result underscores that it matters which evaluation parameters are chosen. If the overarching goal is to decrease health risks, sublethal doses could be useful as they protect human hosts even when mosquito mortality is null.

## 1. Introduction

Despite decades of intensive efforts on multiple fronts, there is no clear end in sight in the difficult battle to control vector-borne diseases (VBDs) [1].

VBDs are having increasingly severe impacts on humans living in tropical, subtropical, and temperate zones [2,3]. The Asian tiger mosquito, *Aedes albopictus* (Skuse, 1894), naturally occurs in Asia and is a vector species that has spread across the globe to Africa, Europe, Australia, the Americas, and the Middle East [4,5,6]. It is a largely anthropophilic and exophilic species that bites hosts during the day [5,7]. With the mosquito’s arrival in novel habitats and increased levels of outdoor activities by humans [8,9,10], there is an evident need to develop both alternatives to topical repellents and customised vector control programmes so that the general public is better protected.

The use of public health pesticides (i.e., household and professional insecticides) has been the main way to control vectors in environments where humans are found [11,12]. In recent years, progress has been made in the design and implementation of new methods that complement chemical control techniques within integrated vector management (IVM) programmes. Such methods may be biological, genetic, or physical in nature [1,13]. However, worldwide, one of the most broadly employed strategies remains chemical control, including insecticide use [14]. That said, no new classes of insecticides have been registered for use in vector control over the last few decades. Indeed, the number of active substances (ASs) that can be employed in insecticidal products has drastically declined because regulatory authorities now give greater weight to concerns about resistance, toxicity, and environmental risks; furthermore, the cost of developing and registering new substances has climbed [15,16].

Household insecticide products are most commonly applied via electrical devices, aerosols, coils, and passive evaporators (e.g., with AS-impregnated papers or plastics) [17,18,19]. Certain volatile insecticides, such as some pyrethroids, can act as either insecticides or repellents depending on the dose, area/volume treated, and/or exposure time [20,21]. Space repellents (SRs) or area repellents (ARs) are chemicals that are applied in vapour form within a given area to prevent contact between vectors and their human hosts. Arthropods are thus deterred from entering spaces occupied by humans, eliminating or reducing the risk of disease transmission. This approach contrasts with the conventional use of chemical insecticides, in which the same ASs are used at higher doses to kill insects [16].

Although it has been recognised for decades that modifying vector behaviour is an important part of disease control efforts, this approach has largely been underutilised [20]. In the context described above, certain volatile insecticides could theoretically be employed as SRs, representing a new paradigm for controlling disease vectors and protecting human hosts. Some examples of molecules that can function as either insecticides or SRs are metofluthrin [19], prallethrin [16], d-allethrin [20], and transfluthrin (TFT) [21,22]. The latter was used in this study.

In most regulatory guidelines, including those used in the European Union (EU), the SRs in insecticides are evaluated based on levels of mosquito knockdown and mortality rather than on the degree of repellence or biting. In contrast, World Health Organisation (WHO) guidelines use the latter metrics when testing the efficacy of SRs in household products [22,23]. In the EU, ASs must be officially registered as both insecticides and repellents if they are to be marketed for each purpose. At present, only geraniol (CAS number 106-24-1) and *Chrysanthemum cinerariaefolium* extract (CAS number 89997-63-7) [16] are authorised for both uses.

To explore how SRs could be used as complementary public health tools, it is recommended that two key directions be pursued. First, for the sake of the general public, there is a need to develop safe and effective alternatives to insecticides and/or topical repellents that provide similar levels of protection both indoors and outdoors [17]. Second, to improve vector control programmes worldwide, it should be determined how SR usage can accompany more classical control strategies [24,25]. A starting point in this work is to recognise and implement new evaluation parameters and/or to make use of ASs that are already registered. One advantage of this approach is that it avoids the prohibitively high cost of developing and registering new ASs or AS uses [16].

This study evaluated the effectiveness of a synthetic type I pyrethroid, transfluthrin. In the EU, TFT is registered for use exclusively as an insecticide. Three approaches were taken. First, an SR method was employed to evaluate the degree of human protection provided by a given dose of spatially applied TFT. Second, a conventional insecticide evaluation method was utilised to examine how the same TFT dose and application method affected mosquito knockdown and mortality. Both experiments were carried out under field conditions where *Aedes albopictus* was the study species. Third, modelling was used to assess the toxicological risks presented by TFT for human and environmental health.

## 2. Results

### 2.1. Assessment of Human Protection

Mean temperature and relative humidity were 27.6 °C (range: 23.4–30.5 °C) and 55.3% (range: 50.5–58.5%), respectively. There was a negative correlation between landing number and temperature, meaning fewer landings occurred at higher temperatures (Pearson correlation: −0.251, *p* < 0.005). In contrast, landing number and relative humidity were not associated (Pearson correlation: 0.109, *p* = 0.209).

The TFT treatment affected landing number (t = 6.54, *p* < 0.00001), which differed both between the control and treatment plots (control vs. treatment zone: *p* < 0.00001 and control vs. treatment-adjacent zone: *p* < 0.00001) and between the treatment zone and the treatment-adjacent zone (*p* < 0.0005) (Figure 1). In line with this result, the percent protection differed between the treatment zone and the treatment-adjacent zone (t = −5.29, *p* < 0.00001; Figure 2).

There was also an influence of sampling time on landing number (t = 4.96, *p* < 0.00001; Figure 2), where landings increased over time. There was a significant interaction between sampling time and treatment (t = −4.37; *p* < 0.0001): the number of mosquito landings increased more rapidly over time in the treatment-adjacent zone than in the treatment zone (Figure 2). Not surprisingly, as time went by, the percent protection declined (t = −5.47, *p* < 0.00001; Figure 3). The pattern was the same in both the treatment zone and the treatment-adjacent zone (t = −1.548; *p* = 0.124), although protection tended to decrease more slowly in the former.

### 2.2. Assessment of Mosquito Mortality

For the first batch of mosquitoes, knockdown (KD) was 100% among the treatment mosquitoes at 15 min; mortality was 97.5% ± 5.95 after 24 h. No KD or mortality was observed among the control mosquitoes. Consequently, mortality differed significantly between the two groups (control vs. treatment: F_6,1_ = 3733.6, *p* < 0.00001). For the second batch of mosquitoes, used to test for the presence of residual effects, KD was null for both the treatment and control groups; the same was true for mortality after 24 h (Figure 3).

### 2.3. Assessment of Human and Environmental Health Risks

Risk modelling found that there were no unacceptable human health risks resulting from primary or secondary exposure to TFT (Table 1).

These results show that the dose of TFT used in the experiment could be safely employed in aerosol form in outdoor settings. For the four scenarios analysed, the PEC/PNEC ratio was less than 1.

Similarly, the assessment of risks to urban and rural environmental health found that the TFT dose was safe given the above intended usage (Table 2 and Table 3). For all the target compartments, the PEC/PNEC ratio was less than 1.

The use of biocidal products outdoors can put insectivorous and/or vermivorous mammals at risk of secondary poisoning (Table 4).

For both groups, the PEC/PNEC ratio was less than 1, indicating there was no risk of secondary poisoning.

## 3. Discussion

In the first experiment, conducted in the field, a sublethal TFT dose applied as an SR reduced *Ae. albopictus* landings and provided a high degree of protection to study participants (>80% for up to 5 h). In the treatment-adjacent zone, mean percent protection declined more quickly (<60% at 3 h). Interestingly, these results were not the consequence of mosquito mortality because, in the second experiment, mortality was null in the treatment gazebos at 1 h post application. Taken together, these findings show that, under outdoor conditions, sublethal doses of TFT may persist in the air or on the ground. While these residual amounts of TFT may not knockdown or kill mosquitoes, they can still provide protection to human hosts within an established area. This effect may result from pyrethroid-induced neuronal hyperexcitation, which can occur at levels much lower than those provoking knockdown and/or mortality [26].

In general, SRs are highly volatile compounds that aerially diffuse within treated areas [27]. Effective examples include volatile pyrethroids such as such as TFT [28,29], metofluthrin [26,30], allethrin [31,32], and prallethrin [17,33]. Indeed, volatile pyrethroids are the most studied type of SRs. Laboratory trials have found that these molecules can reduce landing and biting frequency by 70–100% in the presence of hosts [26,34]. In semi-field trials employing lab-reared mosquitoes and methods informed by WHO guidelines [22], SRs significantly affected mosquito entries, exits, and feeding behaviour in the testing area by reducing mosquito presence and landing rate [31,35]. Field experiments exploiting wild mosquito populations have also found that SRs can greatly impact the degree of mosquito repellence, landing number, and biting behaviour (protection > 70%) [26,32,36].

SRs can be employed in a variety of devices. They can be actively emitted using coils or electric diffusers [18], or they can be passively emitted from a surface impregnated with the AS [26,37,38]. A commonality of the previously cited studies is that they diffused volatile pyrethroids directly into the air. Such methodology is only possible with ASs that display lower vapour pressure than conventional pyrethroids because they can naturally evaporate at ambient temperatures; there is no need for an external energy source (i.e., burning or heating). Examples of such ASs include metofluthrin and TFT. In this study, a combination of passive and active techniques were used. TFT is characterised by low vapour pressure and was actively delivered using a spray. As a consequence, the formula could be easily applied to a surface (i.e. ground) whence it would evaporate slowly into the air for an extended period of time. Although ASs can be released in a variety of ways, the objective always remains the same: to prevent contacts between hosts and vectors by disrupting host-seeking behaviour and reducing or eliminating vector presence within a specific area [22,25].

As noted above, in the first experiment of the study, host protection lasted longer in the treatment zone (>80% at 5 h) than in the treatment-adjacent zone (˂60% at 3 h). Air flow patterns could help explain these results. Because the treatment was applied along the plot perimeter, the plot centre was found within a “bubble” of protection; such was not the case for the treatment the treatment-adjacent zone. In a 2014 study also using TFT (albeit delivered via coils rather than spray), similar findings were obtained: the treatment was more effective when two coils were placed on either side of the study participant, creating a “bubble” of protection, rather than when only one coil was used (i.e., acting as a single “point source”) [39]. Therefore, when developing an SR application approach, the treatment method merits careful consideration. For example, should the product be applied to one-point, multiple points, or along an entire perimeter? Indeed, the choice of method is especially important if products are to be used outdoors, where conditions are less manageable (e.g., more greatly affected by climatic variables). AS effectiveness is shaped by other factors: the greater the distance from the source of emissions, the lower the compound concentration and percent protection [40,41]. The shorter duration of protection in the treatment-adjacent zone may have been affected by both the choice of application method (i.e., along the plot’s perimeter) and the distance between the application location and the point of measurement.

SRs could be exploited to provide protection to the general public in situations where topical repellents currently provide the only recourse against vector-related risks, notably in outdoor settings. In this regards, the WHO has guidelines for evaluating the SRs used in household insecticide products: the focus is placed not on how well volatile pyrethroids provoke knockdown and mortality, but rather on how well they repel insects and inhibit biting [22]. Published in 2013, these guidelines state that spatial repellency is “*a range of insect behaviours induced by airborne chemicals that result in a reduction in human–vector contact and therefore personal protection. The behaviours can include movement away from a chemical stimulus, interference with host detection (attraction inhibition), and feeding response” [22]*. However, the EU has its own guidelines for regulatory assessments of the efficacy of insecticidal and repellent compounds [42]; they include descriptions of the parameters, efficacy criteria, and methodologies to be utilised with different treatment types (e.g., direct spraying, surface treatment, spatial treatment). At present, EU authorities do not officially allow insecticidal compounds to be used as SRs. Because most SRs use volatile pyrethroids, SR evaluation and authorisation within the EU is based on two key parameters—knockdown and mortality. Thus, to fully benefit from the advantages of SRs, the EU must formally accept strategies based on behaviour modification and employ mosquito biting behaviour instead of toxicity/lethality as the basis for evaluation [17,43]. The WHO guidelines could serve as a good starting point for constructing a single set of global standards that allow for alternative ways of deploying existing insecticidal compounds for household use.

Since public health pesticides are employed in proximity to human beings and in sensitive ecological areas, it is essential to properly manage their use [11]. Indeed, biocidal products may have a tremendous impact on humans, animals (including non-target insects) and environmental health [44]. In this context, the EU Commission, as part of rolling out the European Green Deal, has announced two pesticide reduction targets to achieve by 2030: (1) a 50% reduction in the quantities of active substances in commercial pesticides and (2) a 50% reduction in the use of more hazardous pesticides [45]. The study presented here did not examine the effects of sublethal TFT doses on non-target insects, an important concern that should be addressed in future research. However, it did show that using sublethal insecticide doses represents a strategy worthy of further exploration: the sublethal dose used did not kill the target insects, mosquitoes, but still afforded a high level of long-lasting protection to humans. Furthermore, this same dose did not present a threat to the health of humans, vermivorous or insectivorous mammals, or the environment (in target compartments—STPs, aquatic habitats, soil, and groundwater). However, if EU evaluation requirements were to be applied (i.e., high knockdown and mortality rates), the SR would only be considered as effective immediately after application. Consequently, the same dose would have to be unnecessarily reapplied every hour to meet EU regulatory standards. Such usage would likely have pronounced impacts on human, animals and environmental health.

In worldwide efforts to fight VBD transmission and protect human health, vector control remains one of the first lines of defence. It is for this reason that research has often focused on the efficacy of current measures and the development of new vector control tools [43,46]. Studies conducted in areas where VBDs are endemic have found that disease-control tools that target resting mosquitoes and indoor-biting mosquitoes, such as insecticide-treated bed nets (ITNs) and/or indoor residual spraying (IRS), may not be enough to eliminate disease, notably when transmission takes place outdoors instead of indoors [47,48,49]. Since the rate at which mosquitoes bite humans is one of the most important factors shaping disease transmission, ASs that interfere with or prevent mosquito feeding behaviour are likely to have an important influence [39,50]. To improve vector management, it is worth further exploring the utility of SR driven behavioural modifications as an alternative to conventional, mortality-based strategies.

There are multiple arguments in favour of this change in approaches, including many that deserve to be explored before SR-based strategies can be globally implemented. First, this study and others have demonstrated that SRs can be highly beneficial: by reducing the rate at which outdoor mosquitoes bite humans, SRs operate on a key factor underlying disease transmission, thereby increasing levels of host protection. Second, SR-based strategies could be deployed in the form of personal protection products or in complement to vector control programmes based on ITN and IRS use, boosting protection for individuals, households, and/or communities via ASs with very low toxicity in mammals [20].

Before SRs can be more broadly employed as a complementary vector control tool [21,22,23,24,25,26,27,28,29,30,31,32,33], it is necessary to conduct additional epidemiological and entomological studies supporting the idea that SR use can reduce disease transmission [51,52]. This need is highlighted by WHO guidelines in their discussion of vector control efficacy trials: *“Phase III studies should be designed around epidemiological endpoints to demonstrate the public health value of the intervention. Entomological outcomes cannot be used on their own for this purpose, although they can be combined with epidemiological outcomes to evaluate a claimed entomological effect” [53]*.

Because of increasing interest in SRs, a certain amount of epidemiological and entomological research has been conducted to explore their use. That said, much more is needed. For example, a field study in Indonesia using placebo versus metofluthrin coils showed that SR use reduced the number of malaria infections by about 50% [51]. Another study, also performed in Indonesia, found preliminary evidence that passive TFT evaporation could reduce malaria transmission [24]. Thus, at present, more large-scale studies are needed to clarify whether SRs can reduce VBDs [51]. Because conducting such studies is challenging, groups like Ten Bosch et al. (2020) are developing models in which small-scale experimental data can be exploited to predict the epidemiological impacts of SRs at broader scales [54].

Additionally, other research directions should be explored, including mosquito resistance. This study demonstrated that sublethal doses of TFT could protect humans without visual effects on mosquitoes (i.e., knockdown and/mortality 1 hr post treatment). However, it did not examine whether SR exposure could promote the emergence of resistance following continuous exposure. Past work suggests this question is an important one. An in vitro study discovered that when *Aedes aegypti* were repelled by TFT vapours during the F0 generation, their ninth-generation descendants continued to be repelled; however, F0 mosquitoes that were not repelled produced descendants that were insensitive to TFT after four generations [55]. Before SRs are used for repellence at broader scales, it is crucial to explore the potential for changes in vector sensitivity and/or susceptibility to assess the likelihood of resistance [56,57].

Another issue worth examining is the spread of mosquitoes from protected areas to unprotected areas. A study in Tanzania found that houses equipped with TFT-diffusing coils had significantly lower blood feeding rates than houses without coils, which suggests that pyrethroid-based SRs may offer some degree of protection [58]. That said, it remains unknown whether there would be there would be overall benefits for public health, given that harm could result if SR use diverts vectors to non-users. This risk could theoretically be reduced if SR use is combined with push-pull strategies, notably those in which SRs (push) are paired with traps containing CO_2_ or other attractants (pull) [59]. However, several studies have found that such combined approaches do not work better than the use of repellents alone [60,61,62], so further community-level research is merited.

Outdoor-biting vectors are becoming more and more of a concern, and there is a need to incorporate alternative control tools [20,63]. As a result, SRs have the potential to become an important part of vector control programmes and personal protection tool kits. However, much work remains to be done, and it is incumbent upon the international vector control community (e.g., regulatory authorities, industrial stakeholders, and researchers) to confront this challenge by embracing promising innovations. This process will involve making optimal use of available tools and developing new vector control approaches, products, and programmes with a greater view to reducing public health risks. It is essential to promote a shift in current vector control programmes and the approach to regulating the insecticides available to consumers. One key change that could be implemented is to move from killing insects to repelling insects, namely by exploiting innovative products that utilise already approved compounds in novel ways. Ultimately, it is crucial to remember the greater objective: creating vector-free spaces where people are better protected from bites and/or disease transmission while limiting human and environmental health risks.

## 4. Materials and Methods

The study took place in September 2018 in a green landscaped zone within a hospital complex (45°23′58.7″ N, 11°50′31.3″ E) located to the southwest of Padua (Veneto region, Italy). Several studies focused on pathogen screening and insecticide efficacy have been conducted in this area because of its high abundance of *Ae. albopictus* [64,65,66,67]. It was found that *Ae. albopictus* is the only human-biting mosquito active at this location during the day [66]. The study zone contained buildings with courtyard-like patios, which naturally formed independent “experimental plots”. The vegetation included grass; flowering plants, hedges, and bushes (height: 0.5–2 m); ornamental trees (height: 2–5 m); and non-ornamental trees (height: 5–20 m) (Figure 4).

There were three parts to the study. First, two field experiments were performed: (1) an assessment of human protection and (2) an assessment of mosquito mortality. Then, modelling was used to estimate the toxicological risks of TFT for human and environmental health.

### 4.1. Assessment of Human Protection

Ten people (5 men and 5 women) participated in the first experiment. Participants were fully informed about the nature and purposes of the study and about any physical consequences that could foreseeably result from having taken part. Non-smokers were preferentially recruited; if participants were smokers, they were asked to refrain from using tobacco during the testing process and for 12 h prior. Participants were similarly asked to avoid alcohol consumption and the use of perfumes, body lotions, soap, and/or repellents [68]. In addition to these selection criteria, participants were chosen based on their ability to morphologically identify free-flying *Ae. albopictus*.

The degree of protection provided against mosquito bites was quantified using the landing rate count (LRC) method, which quantifies the number of landings that take place during a fixed period of exposure. A landing was defined in the following way: after a mosquito alights on a human, it probes the skin with its proboscis. The detailed methodology is described elsewhere [67].

Over the course of the experiment, temperature and humidity were measured hourly using a portable digital weather station (TFA Dostmann) and a digital thermohygrometer (Lafayette TM-4).

The experimental trials took place from 9:00 to 16:00. Two plots were randomly assigned to each volunteer: one was the treatment plot, and the other was the control plot. The plots measured approximately 3 × 3 m and were separated by at least 20 m to avoid concurrent attraction among plots, as recommended in WHO guidelines [69]. In total, twenty plots were used (Figure 5).

A pre-treatment trial was conducted from 9:00 to 10:00. Three check-ins were performed at twenty-minute intervals. Participants stood for 5 min in the middle of their plot with the lower half of both their legs exposed (i.e., from knee to ankle). During this period, the number of mosquito landings was recorded. After the pre-treatment trial, 10 plots were treated (i.e., the treatment plots), and 10 were left untreated (i.e., the control plots).

The formula used in the study was prepared in Henkel’s chemical laboratory in San Marino. It was water based and contained 0.104%- TFTas the AS (98.5% technical grade, CAS number 118712-89-3) In the field, the formula was applied with an aerosol can that was energetically shaken to ensure maximum homogenisation before application. The perimeter of the plot was then treated with 12.3 ± 0.3 g of formula, which was sprayed directly on the ground within a 10-cm^2^ band along a total area of 9 m^2^. The total surface area treated was approximately 1.2 m^2^.

Following the treatment, study participants stood in each of their two assigned plots (treatment and control) for 5 min once per hour, and the same methodology was used as during the pre-treatment trial. During each of these 5-min periods, the numbers of mosquito landings were recorded as the study participant stood (1) in the centre of the control plot; (2) in the centre of the treatment plot (i.e., the treatment zone); and (3) in a location 1 m from the perimeter of the treatment plot (i.e., the treatment-adjacent zone). The placement of the treatment-adjacent zone was chosen based on the results of previous studies, in which a repellent was located 1 to 1.2 m from human hosts [26,39,40]. The trial series ended when mean percent protection fell below 80% in the treatment zone.

Percent protection was defined as the percent reduction in landings for each participant in the treatment versus the control plot and was calculated as follows (based on Mulla et al. 1971):% Reduction = [100 − (C1/T1 × T2/C2) × 100](1)
where 

C1 = number of mosquito landings in the control plot before treatment

T1 = number of mosquito landings in the treatment plot before treatment

C2 = number of mosquito landings in the control plot after treatment

T2 = number of mosquito landings in the treatment plot after treatment

Although this equation was originally used with estimates of insect larva abundance, it can be applied to any standardised counts that are obtained in treatment and control areas [70].

### 4.2. Assessment of Mosquito Mortality

The second experiment was performed in gazebos (3 × 3 × 3 m) that were open on two sides (Figure 6). The PVC outer cover was UV-stabilized and waterproof. There were four treatment gazebos and four control gazebos. In the treatment gazebos, 12.6 ± 0.7 g of formula was sprayed on the ground around the inner perimeter. The same procedure was used as in the first experiment.

Two batches of mosquitoes were utilised: the first batch was employed immediately after the treatment, and the second batch was employed 1 h post treatment. In each case, three cages of mosquitoes were placed in the gazebos. The cages were positioned 10 cm above the ground. They were cylindrical (diameter: 5.5 cm; height: 5.5 cm) and made of metallic netting (mesh size: 1 mm). Each cage contained 10 non-blood-fed female *Ae. albopictus* that were 5–10 days old. The mosquitoes came from a colony that was maintained at the Entostudio Test Institute (Italy) for the last 10 years. Mosquito-rearing conditions were as follows: temperature of 25 ± 2 °C, relative humidity of 60 ± 5%, and a photoperiod of 12:12 (L:D). To ensure they were active during the experiment, the mosquitoes were given water and 10% sucrose solution ad libitum until the experiment began.

The percent knockdown in the first batch of mosquitoes was measured in the treatment and control gazebos at 15 min and 1 h post treatment. A total of 240 mosquitoes were used: 120 in the treatment group and 120 in the control group. All the mosquitoes were subsequently moved to the laboratory; after 24 h, mortality was evaluated.

At 1 h post treatment, the second batch of mosquitoes was placed in the gazebos to ascertain whether the formula had any residual effects. As above, a total of 240 mosquitoes were employed.

### 4.3. Assessment of Risks to Human and Environmental Health

In the third part of the study, modelling was used to estimate the toxicological risks of TFT for human and environmental health.

#### 4.3.1. Characterising Human Health Risks

This work was carried out using ConsExpo Web (v. 1.0.7; [71]), a tool designed by the Dutch National Institute for Public Health and the Environment (RIVM). In this software, certain parameters can be set to a chosen value, while others are fixed.

As stated in the Materials and Methods section, the TFT-based formula was applied to a total surface area of approximately 1.2 m^2^.

Because ad hoc Human Health Risk Assessment (HHRA) models lack handling the condition of aerosols deployed outdoors, this study performed ConsExpo models for indoor situations assuming worst-case conditions.

HHRAs were carried out for two populations: adults and toddlers (1–2 years old). Two different types of exposure were considered: primary exposure (e.g., direct exposure via formula application) and secondary exposure (i.e., indirect exposure via various mechanisms). Secondary exposure can occur because of lingering formula residues. It was assumed that volatilised residues could be inhaled. It was also assumed that dermal exposure could occur given the presence of residues on local surfaces. Additionally, toddlers were considered to be at risk of oral exposure. Thus, four different scenarios were explored (Table 5).

**Scenario 1:** The risk of primary inhalation exposure was estimated using the pre-pressurised aerosol spray can model [72]. Exposure values for different body parts were chosen based on the 2015 Biocides Human Health Exposure Methodology (BHHEM, p. 220) [73]. The parameters used in the analysis of this scenario are described in the Appendix A.

In the next three scenarios, exposure was secondary. It was assumed that adults and toddlers could inhale volatilised residues and/or come into dermal contact with residues on local surfaces. Moreover, because toddlers frequently engage in hand-to-mouth (HTM) contact, they were also at risk of ingesting the residues.

**Scenario 2:** The risk of secondary exposure during the post-application period was estimated using the exposure to vapour: an evaporation model was used, derived from the Pest Control Products Fact Sheet [74]. As mentioned above, exposure was assumed to take place indoors because such would be the worst-case situation and would thus also extend to outdoor exposure.

Along these lines, it was also assumed that individuals would remain in the treated area for 12 h following formula application. The default parameter values were modified to incorporate recommendation no. 14 of the Biocidal Products Committee (BPC) Ad hoc Working Group on Human Exposure in relation to body mass (adults: 60 kg and toddlers: 10 kg) and inhalation rate (adults: 16 m^3^/d and toddlers: 8 m^3^/d) [75]. The parameters used in Scenario 2 are described in Appendix A.

**Scenario 3:** Secondary exposure can occur during the post-application period when people come into dermal contact with residues via their bare hands, feet, arms, and legs. It was considered that 100% of the skin on the above body parts was contaminated upon contact with residues on local surfaces, which is a highly conservative assumption. The parameters used in Scenario 3 are described in Appendix A.

**Scenario 4:** Toddlers exhibit a great deal of HTM contact. Therefore, a percentage of any residues transferred to their hands will be dislodged by saliva and eventually ingested, leading to oral exposure. The parameters used in Scenario 4 are described in Appendix A.

#### 4.3.2. Characterising Environmental Health Risks

Environmental health risks were assessed using three ECHA publications: the 2008 Product Type 18 Emission Scenario Document (ESD) [76]; the 2017 Guidance on Biocidal Products Regulation (BPR; Volume IV, Environment – Assessment and Evaluation, Parts B + C) [77]; and the 2018 Technical Agreements for Biocides (TAB) [78].

This process employed the predicted no-effect concentrations (PNECs) from the TFT assessment report (RMS, Netherlands, 2014 [79]) and the Bayer report—Relevant endpoints and PNEC derivation Environment & Ecotoxicity (TFL-PAI-2018-v1) [80]. Calculations were performed in Excel using equations from the above 2017 Guidance on BPR [77].

Risks were assessed for two types of environments: rural areas and urban areas. In rural areas, biocidal products applied to the ground result in direct exposure to the soil compartment. Following the first rainfall event, the products end up on unpaved soil. The environmental compartments facing exposure are thus aquatic habitats (water and sediment), soil, and groundwater.

The situation is different in urban areas. According to the ESD [76], biocidal products applied to the ground are most likely to end up on non-permeable substrates (e.g., pavement, concrete, or asphalt). Then, after the first rainfall event, the products flow across such surfaces to storm water drainage systems. Consequently, sewage treatment plants (STPs) represent the main compartment facing exposure.

For each environmental compartment, risk was characterised using the ratio of predicted environmental concentrations (PECs) to predicted no-effect concentrations (PNECs).

Additionally, the risk of secondary poisoning was calculated for mammals. Indeed, mammals may be indirectly exposed because they can consume contaminated insects and/or earthworms. AS concentrations in earthworms were calculated using the 2017 Guidance on BPR [77].

For insectivorous species, estimated theoretical exposure (ETE) was calculated using estimated daily intake and was expressed in PEC_oral_ per day (mg AS/kg body mass of the prey species/day over mg AS/kg food/day).
ETE = (FIR/BM) × C × AV × PT × PD (mg/kg BM/d)(2)
where

FIR = rate of food intake for insectivorous or vermivorous species (fresh mass; g/d)

BM = body mass of insectivorous or vermivorous species (g)

C = concentration of the AS in the fresh diet (insects or worms; mg/kg)

AV = avoidance factor (1 = no avoidance, 0 = complete avoidance)

PT = fraction of the diet obtained within the treated area (value between 0 and 1)

PD = fraction of the food type in the diet (value between 0 and 1; one or more types)

### 4.4. Statistical Analysis

R was used to perform all the statistical analyses [81,82,83,84,85,86].

To assess the degree of human protection achieved, generalised linear mixed models (GLMMs) were performed to compare the numbers of mosquito landings (Poisson error distribution and log-link function; MASS package) among the treatment groups (the control plot, the treatment zone, and the treatment-adjacent zone) and sampling times. Participant identity was included as a random factor. When overall significant differences were detected, t-tests employing pooled standard deviation and Bonferroni corrections were used to perform pairwise comparisons. The percent protection seen in the treatment zone versus the treatment-adjacent zone was compared using GLMMs (Gaussian error distribution and identity link function; nlme package) in which participant identity was a random factor.

The relationship between the number of mosquito landings and the two abiotic factors—air temperature and relative humidity—was examined using Pearson’s correlation coefficients.

To assess mosquito mortality, the numbers of dead mosquitoes in the control versus the treatment cages were analysed using one-way analyses of variance (ANOVAs). First, normality and homogeneity of variance were ensured by performing a Kolmogorov-Smirnov’s test and a Levene’s test, respectively.

## Figures and Tables

**Figure 1 pathogens-10-01171-f001:**
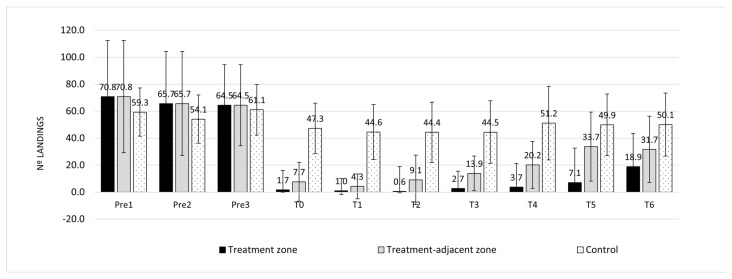
*Aedes albopictus* landings per minute (mean ± SD) during the assessment of human protection. Pre-treatment (Pre). Time in hours after application (T). The treatment groups were as follows: control = participants stood in the centre of a control plot; treatment zone = participants stood in the centre of a treatment plot; and treatment-adjacent zone: participants stood 1 m from the perimeter of a treatment plot. SD, standard deviation.

**Figure 2 pathogens-10-01171-f002:**
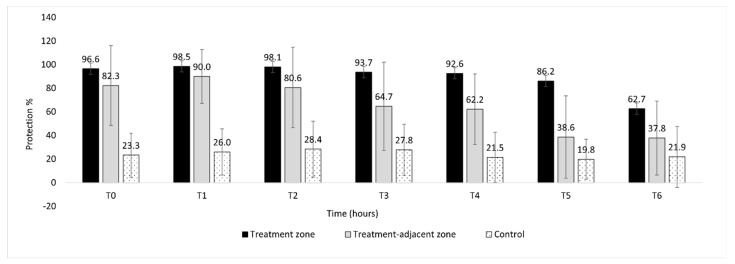
Percent protection (mean ± SD) against *Aedes albopictus* mosquitoes during the assessment of human protection. The treatment groups were as follows: control = participants stood in the centre of a control plot; treatment zone = participants stood in the centre of a treatment plot; and treatment-adjacent zone: participants stood 1 m from the perimeter of a treatment plot.

**Figure 3 pathogens-10-01171-f003:**
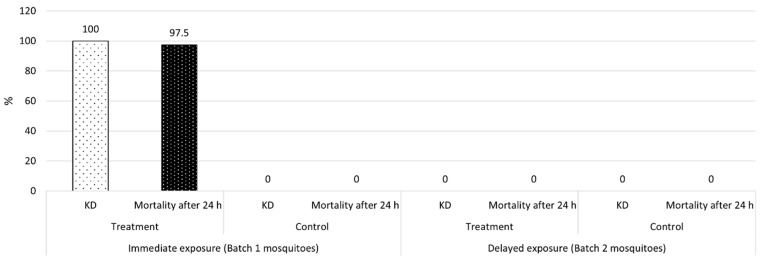
Percent knockdown (KD) and mortality (mean ± SD) in *Aedes albopictus* during the assessment of mosquito mortality. KD was measured at 15 min and 1 h post application for the treatment and control mosquitoes. Mortality was assessed 24 h later.

**Figure 4 pathogens-10-01171-f004:**
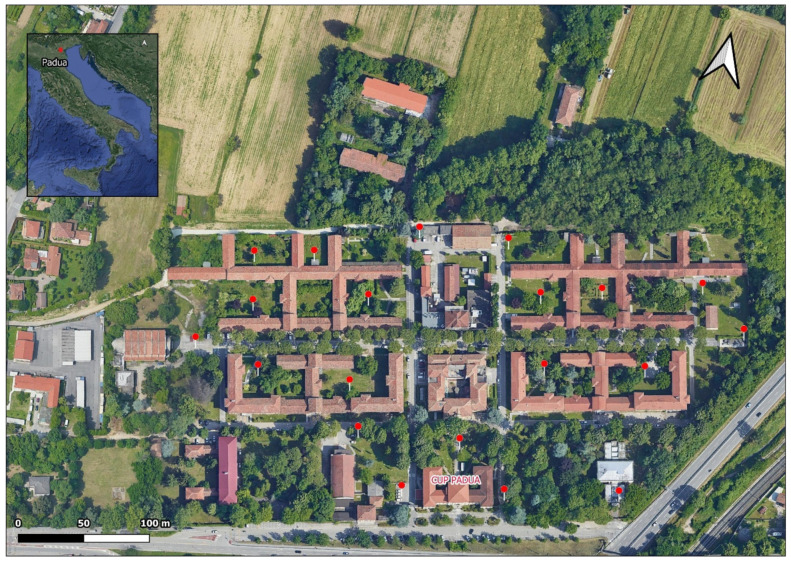
Study zone (45°23′58.7″ N, 11°50′31.3″ E) located to the southwest of Padua, Italy. The red markers indicate the locations of the experimental plots used in the assessment of human protection. Source: Google Earth.

**Figure 5 pathogens-10-01171-f005:**
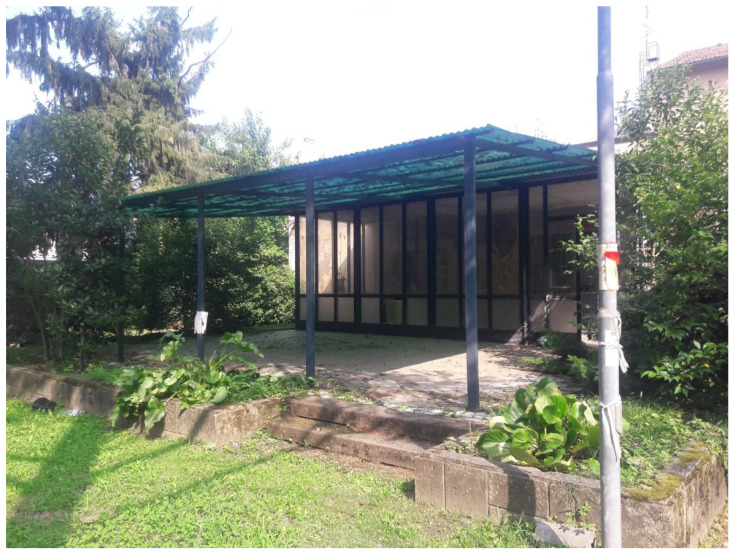
Experimental plot located under a covered patio. The vegetation consisted of ornamental plants, common garden shrubs, and trees.

**Figure 6 pathogens-10-01171-f006:**
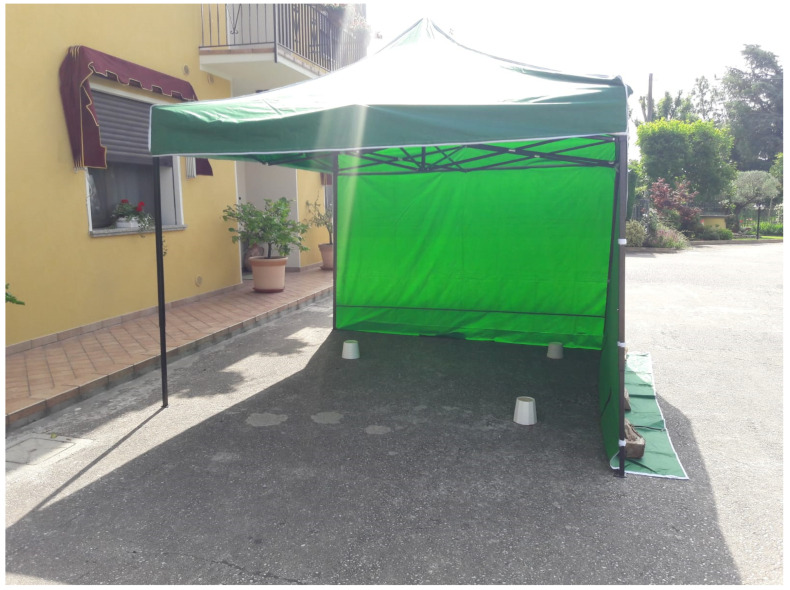
Treatment gazebo located in a patio. The vegetation consisted of ornamental plants, common garden shrubs, and trees.

**Table 1 pathogens-10-01171-t001:** Acceptable exposure levels (AELs) for adults and toddlers in the different HHRA scenarios. Scenario 1—primary exposure via inhalation during spraying; Scenario 2—secondary exposure via inhalation of volatilised residues; Scenario 3—secondary exposure via dermal contact with residues on surfaces; and Scenario 4—secondary exposure via ingestion of residues on surfaces.

Scenario	AEL (mg/kg BM/d)	Estimated Uptake (mg/kg BM/d)	Estimated Uptake/AEL (%)	Acceptable? (Yes/No)
[1] Adults	0.01	2.02 × 10^−4^	2	Yes
[2] Adults	0.01	1.6 × 10^−6^	<1	Yes
[2] Toddlers	0.01	4.80 × 10^−6^	<1	Yes
[3] Adults	0.01	1.33 × 10^−3^	13	Yes
[3] Toddlers	0.01	2.02 × 10^−3^	20	Yes
[4] Toddlers	0.01	4.30 × 10^−4^	5	Yes

**Table 2 pathogens-10-01171-t002:** Estimated predicted environmental concentration (PEC) values for urban and rural environments and the target compartments (i.e., sewage treatment plants [STPs], aquatic habitats [water, sediment], soil, and groundwater).

	Compartments				
	STP	Aquatic Habitats		Soil	Groundwater
Environment	PEC [mg/L]	PEC_water_ [mg/L]	PEC_sediment_ [mg/kg ww *]	PEC [mg/kg ww *]	PEC [μg/L]
Urban	1.14 × 10^−2^	1.06 × 10^−6^	1.16 × 10^−3^	2.65 × 10^−4^	2.99 × 10^−4^
Rural	NA	NA	NA	2.92 × 10^−3^	3.30 × 10^−3^

ww *: wet weight. NA = not applicable.

**Table 3 pathogens-10-01171-t003:** Estimated PEC/PNEC ratios for urban and rural environments and the target compartments (i.e., sewage treatment plants [STPs], aquatic habitats [water, sediment], soil, and groundwater).

	STP	Aquatic Habitats		Soil	Groundwater
Environment	PEC/PNEC	PEC/PNEC_water_	PEC/PNEC_sediment_	PEC/PNEC	PEC/PNEC
Urban	2.00 × 10^−3^	0.61	0.65	3.01 × 10^−3^	NA *
Rural	NA	NA	NA	3.32 × 10^−2^	NA *

* **Groundwater** was considered to be at zero risk because concentrations were below 0.1 µg/L for all scenarios. NA = not applicable.

**Table 4 pathogens-10-01171-t004:** Estimated PEC/PNEC ratios for insectivorous and vermivorous mammals. PNEC, predicted no-effect concentrations.

Exposure Scenario	PEC [mg/kg] *	PEC [mg/kg] *	PEC/PNEC	
	Acute Oral Exposure	Short-Term Oral Exposure	Acute Oral Exposure	Short-Term Oral Exposure
Vermivorous mammals	1.52 × 10^−1^		2.28 × 10^−2^	
Insectivorous mammals	1.01 × 10^−8^	3.70 × 10^−9^	1.52 × 10^−9^	5.54 × 10^−10^

**Table 5 pathogens-10-01171-t005:** Human Health Risk Assessment (HHRA) scenarios in which there were primary or secondary exposures via different routes (inhalation, dermal, or oral) in adults and/or toddlers.

Scenario Number	Scenario Description	Primary or Secondary Exposure—Description	Population
1	Application to outdoor surfaces	Primary exposure Low-pressure spraying of the ready-to-use formula	Adults
2	Inhalation of volatilised residues	Secondary exposure General public exposed during the post-application period via residue inhalation	Adults, toddlers
3	Dermal contact with residues on surfaces	Secondary exposure General public exposed during the post-application period via dermal contact	Adults, toddlers
4	Ingestion of residues on surfaces	Secondary exposure Children exposed during the post-application period via dermal and oral contact (hand-to-mouth behaviour)	Toddlers

## Data Availability

The datasets generated during and/or analysed during the study are available from the corresponding author upon reasonable request.

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
