# Peer review of "To Kill or to Repel Mosquitoes? Exploring Two Strategies for Protecting Humans and Reducing Vector-Borne Disease Risks by Using Pyrethroids as Spatial Repellents"

_pathogens, 2021, doi:10.3390/pathogens10091171_

Round 1

Reviewer 1 Report

1) Both the title and the abstract do not indicate the measure under study, mislead the author by suggesting research on several measures. Especially in the abstract, it is necessary to precisely state what was tested and how. 2) Pages 6-7, lines 220-241: In my opinion, the information about the suggested changes in the regulations does not increase the value of work, nor does it refer to the results obtained. 3) There is no reference in the discussion to the global trend of reducing the concentration of insecticides and their harmful effects on human and animal organisms. 4) In the summary, there is no reference to the tested pyrethroid. Should be redrafted so that the summary relates to the concrete results. 5) Abbreviations in keywords should be expanded and all keywords should be arranged alphabetically. 6) Page 2, lines: 87: technical description of transfluthrin for material and methodology chapter. 7) Figure 1: in my version, the treatment zone and the adjacent zone look similar. 8) Tabel 1: there is no explanation of the abbreviations below the table. 9) Materials and methods, pkt. 4.2: control gazebos are not discribed. 10) Supplementary materials are missing in my version of the article.

Reviewer 2 Report

The manuscript titled “To Kill or to Repel Mosquitoes? Two Strategies for Protecting Humans and Reducing Vector-Borne Disease Risks”. The author’s research study explores how the use spatial repellents to promote public health; recommend new methods for insecticide evaluation. Additionally, identify problems that may arise from the use of spatial repellents. The sample size used was adequate for the experimental design with multiple sites and control areas. The data presented are sound and do justify the author's conclusion, supporting the claims that as outlined in their research objectives. The analysis, tables and graph of the author's results are well simplified and comprehensible. Overall, the manuscript was well prepared and this study can be replicated with the information provided. Proper methods were used, and the data is correctly analyzed, interpreted, and justifies the conclusions. The author’s results from this study demonstrated that the spatial repellent transfluthrin may have the potential to become an important part of a vector control program. This manuscript describes interesting results and important for the development for better protection of humans from biting insects and vector born disease. It is very important to consider the effects on non-target insects and long and short-term human exposure.

Figures 1,2,3:  If this is to be published in a journal, the bars in the graph need to be a bit more contrasting. If online, use different colors.

Line 36: It is a largely anthropophilic and exophilic species that bites bite hosts…

Lines 171, 204: (~60% at 3 h) you have exact data, should be (<60% at 3 h).  There should be no approximations reporting scientific data.

Lines 243-244: What are the effects on non-target insects and long-term exposure of humans?

Lines 270-271: Where do you see this being used?  As a control method for residential backyards or rural homes in developing nations?

Lines 276-279:  This research is a good first step in advancing to your future research goals. The Indonesia study cited would be a good experimental design for testing transfluthrin as a special repellent.

Round 2

Reviewer 1 Report

I recommend the manuscript to be accepted. But I have got one more point to correct: Figure 2 - lack of explanation of T0 do T6 abbreviations.